# Intensive Behavioural and Pharmacological Treatment for Tobacco Dependence in Pregnant Women with Complex Psychosocial Challenges: A Case Report

**DOI:** 10.3390/ijerph17134770

**Published:** 2020-07-02

**Authors:** Melissa A. Jackson, Amanda L. Brown, Amanda L. Baker, Adrian J. Dunlop, Angela Dunford, Gillian S. Gould

**Affiliations:** 1Hunter New England Health Local Health District, Newcastle, New South Wales 2300, Australia; Amanda.Brown@health.nsw.gov.au (A.L.B.); Adrian.Dunlop@health.nsw.gov.au (A.J.D.); 2School of Medicine and Public Health, University of Newcastle, Callaghan, New South Wales 2308, Australia; Amanda.Baker@newcastle.edu.au (A.L.B.); gillian.gould@newcastle.edu.au (G.S.G.); 3John Hunter Hospital, Hunter New England Health Local Health District, New Lambton, New South Wales 2305, Australia; Angela.Dunford@health.nsw.gov.au

**Keywords:** tobacco use, smoking cessation, pregnancy, harm reduction, contingency management, nicotine replacement therapy, counselling, substance use disorders, social disadvantage, mental illness

## Abstract

Up to 95% of women who use other substances also smoke tobacco during pregnancy. Challenging psychosocial circumstances and other barriers that contribute to high levels of tobacco dependence result in few quitting successfully. This case report describes the treatment of a highly tobacco dependent 34-year-old pregnant woman with a history of recent substance use, mental illness and trauma, enrolled in the Incentives to Quit Tobacco in Pregnancy program. Heavy smoking, both during the day and overnight, was reported. An extensive history of quit attempts, as well as a strong desire to cease tobacco use during pregnancy, was also noted. Treatment utilising extensive behavioural supports, including financial incentives for carbon monoxide verified abstinence and telephone-based counselling, in combination with nicotine replacement therapy (NRT), was offered to assist cessation. Excellent uptake and adherence to all aspects of treatment saw tobacco cessation achieved and maintained for 24 weeks while on the program. NRT used at doses well above those recommended for pregnancy was required to alleviate strong withdrawal symptoms and maintain abstinence. Daily monitoring of carbon monoxide, financial incentives for continued abstinence and regular phone support were critical to maintaining motivation and preventing relapse to smoking. Post-program relapse to smoking did occur, as is common, and highlights the need for longer-term intensive support for pregnant women with complex behavioural and social problems. Given the prevalence of tobacco smoking in such populations, long-term harm reduction treatment models using extensive behavioural support in combination with NRT should be considered for inclusion in current smoking cessation guidelines.

## 1. Introduction

The highest prevalence of tobacco smoking among disadvantaged populations occurs in those who report harmful alcohol and other drug use [1]. Pregnant women who use alcohol and other drugs are up to five times more likely to smoke than pregnant women who do not, with prevalence estimates ranging from 71 to 95% [2,3,4]. In addition to the problems caused by tobacco and substance use, women from this group are more likely to encounter socioeconomic disadvantage [5] and have concurrent mental health problems [6] and a history of trauma [7]. The use of tobacco as a coping mechanism to relieve symptoms of stress, depression and anxiety [4,8] and a likelihood of being highly tobacco-dependent [9] can contribute to women who use other substances experiencing more severe nicotine withdrawal and greater difficulty quitting. Despite this, women with substance use concerns report an overwhelming desire to stop smoking tobacco [4,10], although most will continue throughout their pregnancy and beyond [2]. Effective treatments for tobacco dependence that address the psychosocial barriers facing pregnant women with substance use concerns are limited [3].

Nicotine replacement therapy (NRT) is considered a first-line pharmacological treatment for tobacco dependence in Australia and elsewhere [11], although its use in pregnancy has been controversial [12]. While its effectiveness in pregnancy is supported [13], uncertainty about the potential foetal harm caused by nicotine and a lack of confidence by health practitioners in prescribing NRT has restricted recommendations for maternal use [12]. From a harm reduction perspective, “clean nicotine” poses less risk to a pregnant woman and her foetus than exposure to the multitude of toxins ingested when smoking combustible cigarettes [14]. Current guidelines in many countries recommend that pregnant women use only the minimum effective, short-acting (e.g., gum, lozenge, oral spray, inhalator) dose of NRT required to achieve abstinence [13,15]. Such forms and dosages may not adequately relieve strong withdrawal symptoms and might account for the poor smoking outcomes seen in trials of pregnant smokers, as explained in Chamberlain et al. [16]. Combining NRT with behavioural supports (e.g., counselling, financial incentives) is known to improve smoking cessation outcomes [17].

This case report describes the treatment challenges encountered by a pregnant woman with a history of recent substance use enrolled in the Incentives to Quit Tobacco in Pregnancy (iQuiP) program. iQuiP is a comprehensive smoking treatment being piloted across several substance use programs in antenatal clinics in Australia [18]. It is offered to all clinic clients who are identified as smokers and offers recipients a combination of evidence-based behavioural and pharmacological cessation support (details below) to assist smoking cessation or reduction during their pregnancy through to 12 weeks postpartum.

## 2. Case Presentation

### 2.1. Demographics and History

A 34-year-old client of the Substance Use in Pregnancy and Parenting Service (SUPPS) within a New South Wales (NSW) regional tertiary referral hospital was referred to the iQuiP program. Initial antenatal presentation occurred at 23 weeks’ gestation. The woman resided in public housing and was the sole caretaker of a four-year-old daughter, receiving income support for full-time parenting. She reported experiencing physical and emotional violence prior to and during her current pregnancy, perpetrated by the father of her children. This relationship had since ended.

Her medical history indicated asthma and beta thalassemia, a blood disorder that reduces haemoglobin production causing a lack of oxygen to many parts of the body. She had a current diagnosis of Bipolar Affective Disorder, Post-Traumatic Stress Disorder and a previous diagnosis of Generalised Anxiety Disorder. She gave a history of cannabis consumption, commencing in her late 20s, with use of approximately one gram per day mixed with tobacco, primarily during the evening. Cannabis consumption had ceased earlier in the current pregnancy and there was no current use of alcohol, opiates, amphetamines or other substances. Ongoing problems with insomnia and general restlessness were described. The research approval was granted by Hunter New England Human Research Ethics Committee (Reference 17/04/12/4.05).

### 2.2. Tobacco Smoking

The woman reported smoking approximately 30 tailor-made or hand-rolled cigarettes per day. She stated that she was highly motivated to stop smoking, citing the health impacts of smoking on her unborn baby and her own respiratory health. Initial assessment revealed that tobacco smoking commenced at age 11, influenced by exposure to parental smoking and peer pressure. She described waking to smoke approximately six times overnight.

The participant had an extensive history of quit attempts. Her longest period of cessation was aided initially by nicotine patches and lasted four years, with social influences responsible for her resumption of smoking. Other periods of cessation had been assisted by NRT and behavioural support including hypnosis, counselling and Quitline services. Her most recent attempts to cease tobacco had been in response to surgery and health concerns, although no attempts to cease during pregnancy had been made.

An extended interview to understand the participant’s tobacco smoking-related experiences revealed strong feelings of guilt about smoking during pregnancy and its adverse impact on her unborn child. She reported smoking throughout her first pregnancy and that her daughter’s low birth weight had been attributed to smoking. She described strong associations between smoking and coffee consumption, and also boredom. At home, smoking was permitted inside some rooms, as well as inside the car when children were not present. Her significant social supports (mother and sister) were daily tobacco smokers. An intense fear of quitting and of the associated withdrawal symptoms were highlighted. Despite this, the participant expressed a strong desire to cease tobacco for the long-term.

### 2.3. Laboratory and Diagnostic Data

Baseline data collection included an evaluation of current depression (Patient Health Questionnaire; PHQ-9) [19], anxiety (Generalised Anxiety Disorder; GAD-7) [20], alcohol consumption (Alcohol Use Disorders Identification Test for Consumption; AUDIT-C) [21], substance use and wellbeing (Australian Treatment Outcomes Profile; ATOP) [22]. A screen for early exposure to physical, sexual and/or emotional abuse and neglect (Childhood Trauma Questionnaire; CTQ) [23] was also undertaken (see Appendix A for instrument scoring details). Scores placed the participant in the severe range for depressive symptoms (score 16/27) and the moderate range for anxiety symptoms (score 10/21). No past-month use of alcohol, cannabis, methamphetamine, opioids, benzodiazepines or cocaine were reported. On a visual analogue scale of 1–10 (where 1 is poor and 10 is excellent), the participant self-rated her emotional health as 7, physical health as 4 and overall quality of life as 7. Childhood trauma scores suggested severe to extreme maltreatment in the categories of physical, sexual and emotional abuse and emotional neglect. Physical neglect fell within the moderate to severe category.

At baseline, a Fagerström Test for Nicotine Dependence [24] score of 9 determined heavy nicotine dependence. Carbon monoxide (CO) concentrations in the exhaled breath indicated 44 parts per million (ppm). To assess nicotine uptake from NRT during treatment, urine cotinine levels were sampled during week 4 of abstinence.

### 2.4. Medication History 

Her Bipolar Affective Disorder was currently being treated with olanzapine (10 mg daily) and escitalopram (20 mg daily). Lithium treatment had ceased early in the pregnancy. Asthma was treated with salbutamol and fluticasone metered dose inhalers, as required. Previous attempts at tobacco cessation included using NRT (both short- and long-acting), varenicline and e-cigarettes. 

### 2.5. Progress Through the Program

The intervention offered by iQuiP provided financial incentives for CO verified smoking abstinence or reduction, an individually tailored supply of NRT and telephone-based counselling support. Incentives have been cited as the single most effective treatment for pregnant women to stop smoking [16]. CO was self-monitored and captured by video using a monitor provided by the study. CO samples were recorded and uploaded to the study team twice daily (a minimum of eight hours apart) for the first four weeks of abstinence, then once daily thereafter. Each sample below the abstinence threshold of 6 ppm earned a financial incentive. Incentive dollar values increased with every verified sample and were available from program enrolment to delivery day. All short- and long-acting forms of NRT were available and supplied free of charge to participants and their partners who smoked and were available from program enrolment until 12 weeks postpartum. A telephone-based counselling program designed for the current study was available for the duration of the program until 12 weeks postpartum, with the goal of providing contact as often as required by the participant.

At the baseline assessment, where initial data collection and CO monitoring training were completed, an NRT sample pack containing 24-hour 21 mg patches, 4 mg gum, 15 mg inhalators, 1 mg oral spray and 4 mg lozenges was provided, along with education on their correct use. The participant was encouraged to try each of these nicotine formulations over the following days to assess suitability and effectiveness, with favoured products then supplied as needed for the duration of the program. Due to overnight smoking and in liaison with the antenatal service, the participant was advised to wear a patch for 24 hours. Weekly assessments by telephone were taken thereafter to monitor cigarette consumption, review CO monitoring and incentives and track NRT adherence and side-effects.

Abstinence was achieved eight days after program enrolment and intense cravings and urges to smoke were reported during the first two to three weeks. A combination of patch and several short-acting NRT products helped achieve abstinence, with the patch and oral spray particularly useful for the discontinuation of overnight smoking. Consumption of all provided forms of NRT increased over the following two weeks. By week 3 of abstinence, daily usage was self-reported as: 1 × 21 mg 24-hour patch, 6 × 4 mg gum, 7 × 4 mg lozenges, 6 × 15 mg inhalator cartridges and 43 × 1 mg sprays of oral spray, equating to approximately 180 mg of nicotine consumption over a 24-hour period. 

While it is recognised that nicotine absorption from NRT is slower than that of inhaled nicotine, and that a portion of oral nicotine will be swallowed and inactivated [25], a urinary cotinine sample was completed to assess the risk of excessive nicotine dosing. Cotinine is a proximate metabolite of nicotine and used as a biomarker of nicotine exposure. It remains stable within the body, having an average half-life during pregnancy of approximately nine hours [26]. Sample results indicated a nicotine exposure of 2041 µg/litre. Typical peak cotinine concentrations of a daily smoker range from 1000 to 8000 µg/litre [27], placing the participant’s exposure at the lower end of nicotine consumption. Over the following weeks, NRT consumption stabilised and daily use of a 24-hour patch and approximately 90 sprays at 1 mg/spray of oral spray were reported by week 6 of abstinence.

Additional behavioural supports helped maintain abstinence for a period of 12 weeks until delivery of a baby boy weighing 3300 grams at 38.5 weeks gestation. The positive reinforcement of cessation using financial incentives provided much-needed motivation, with 101 verified CO samples submitted indicating breath CO < 6 ppm. During this period, the participant accumulated $1069.90 in voucher-based incentives. Ten 30-minute sessions of telephone-based support were also completed.

Postpartum abstinence was maintained—assisted by NRT and two sessions of telephone support—despite financial incentives ceasing at delivery. Additional telephone support calls were offered during this period, but not utilised. At the 12-week postpartum follow-up, abstinence was verified with breath CO of 0 ppm. Whilst the use of nicotine patches had ceased due to problems with poor adhesion to moist skin during the heat of summer, the use of oral spray was maintained at approximately 90 sprays per day. Consideration was given to the barriers of maintaining this level of use post-program, given its price and lack of subsidised access. A recommendation to reconsider patches or alternative forms, such as gum or lozenges, was made. These are subsidised by the Pharmaceutical Benefits Scheme for a once-yearly 12-week supply, although a script was not requested or provided. 

Program participation ceased at 12 weeks postpartum. The participant’s ongoing care and support was transferred to a community health-based substance use counselling service. Follow-up reports indicated that in the months following discharge, the participant returned to both cigarette smoking and substance use. Documented challenges and contributing factors to relapse indicated low mood, loneliness and unstructured spare time at home during the day, as well as an inability to afford regular purchases of the oral spray.

## 3. Discussion

This case presentation draws attention to the challenges of smoking cessation faced by pregnant women with complex behavioural and social problems. We described a 34-year-old antenatal client with a history of substance use, mental illness and trauma, who required extensive support to achieve and maintain abstinence from tobacco smoking for 24 weeks while on the iQuiP program. Strong withdrawal symptoms associated with heavy tobacco dependence, concomitant cessation of substance use [28], overnight tobacco smoking [29] and accelerated nicotine metabolism [30] required intensive behavioural support and high doses of NRT to overcome. Unique insights into the participant’s dependence and cessation journey through regularly monitored biofeedback, objective assessments of nicotine exposure and detailed case notes were highlighted.

The profile of the case presented is broadly representative of women who attend the high-risk antenatal clinic. Women are referred from general antenatal or medical services for support with alcohol and other drug concerns (not including tobacco). A retrospective audit of clinic records (N = 75) completed in 2017 suggested that cannabis was the most commonly used substance (41%), followed by opioids (22%) and amphetamines (20%). Of women attending, 92% smoked cigarettes during their pregnancy, with 36% of these self-reporting a reduction in cigarettes smoked per day and 11% quitting [31]. Significant co-morbid mental health and social issues, including involvement with government family and justice services, are highly prevalent. The audit also indicated that the average gestational age for first antenatal service appointment was 25 weeks, aligning with our case study’s presentation at 23 weeks. In Australia, women with substance use concerns are more likely to present later to antenatal care services than pregnant women in the general population, or to not receive care at all, with stigma often being a major barrier to accessing support [32]. Late attendance hampers treatment outcomes and prolongs neonatal exposure to potentially harmful toxins. The broader need for strategies to identify and engage at-risk women earlier in the prenatal period is acknowledged.

The NRT dose consumed by this participant was well above that recommended for pregnancy but was required to maintain tobacco abstinence. The iQuiP treatment regimen and decision-making of associated clinicians was guided by an overarching philosophy of tobacco harm reduction. The regimen encouraged the use of enough NRT to avoid severe nicotine withdrawal symptoms and allow the participant to focus on the behavioural aspects of cessation, in contrast to the approach of many antenatal healthcare settings where minimal exposure to nicotine is considered the best solution for optimal maternal and foetal health outcomes.

Despite what appeared to be an excessive consumption of nicotine via NRT use, urinalysis indicated that cotinine levels were similar to those of people consuming low levels of tobacco daily. It should be noted that levels of cotinine may be three times lower in pregnant smokers than non-pregnant smokers, due to increased clearance rates [32]. Thus, cotinine levels may not be representative of nicotine exposure [33], but are nevertheless valuable in assessing the risk of excessive nicotine dosing. Moreover, these results highlight the risk of NRT under-dosing and potential impairment of a quit attempt. Recommendations for higher doses to assist maternal cessation, especially in highly dependent smokers, have been made in light of inconsistent evidence for NRT in pregnancy [16]. The current case provides preliminary evidence for this.

Furthermore, the case highlights the need for a long-term government subsidised supply of NRT, supported by behavioural therapy, to assist and maintain abstinence in heavily dependent smokers. Free NRT played a pivotal role in sustaining abstinence in this case; for those without access to a funded program, the current scheme subsidised by the Australian Government of a once-per-annum 12 week supply for non-indigenous Australians would not adequately assist them to achieve cessation. This is further compounded by the lack of subsidised combination NRT therapy (patch plus short-acting form) or fast-acting varieties such as oral spray, both of which are effective and well-suited to highly dependent smokers [11,34].

There is also a need for smoking cessation programs such as iQuiP to provide longer-term intensive behavioural support to assist heavily dependent smokers in remaining smoke free in the postpartum period and beyond. Relapse is common in the substance dependence cycle, and tobacco dependence is also a chronic relapsing condition. Most (75%) of women who quit during the prenatal period will return to tobacco smoking within 12 months of delivery [35]. Relapse is mediated by multiple environmental and individual factors, including stress and depression [34]. In this case, the termination of incentives at birth, a reduction in telephone support, the discontinuation of free NRT and the prohibitive price of oral spray in particular, may have also contributed to relapse. Continued intensive support, particularly contingent incentives and more frequent counselling, could provide positive reinforcement and is well placed to motivate women during challenging circumstances when the potential for relapse is high. This, combined with a subsidised ongoing supply of NRT, could have ultimately made the difference between relapsing and maintaining long-term abstinence from tobacco. Additional long-term follow-up with participants would potentially supply a rich source of information to inform future approaches.

## 4. Conclusions

The case highlights the complexities of treating tobacco smoking in pregnant women with complex behavioural and social challenges and the need for intensive supports to achieve abstinence. It appears that the comprehensive multimodal approach provided in this case may benefit pregnant women who engage in high-risk behaviours such as substance use to change smoking behaviours in the prenatal period. Given the elevated prevalence of tobacco smoking in high-priority maternal populations, treatment models using intensive behavioural support in combination with a long-term supply of NRT could be considered for inclusion in smoking cessation guidelines for all such groups. These include women who are substance dependent, homeless, mentally unwell, socially disadvantaged and potentially includes women who are Aboriginal and Torres Strait Islanders, depending on community acceptance.

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
