# Peer review of "Intensive Behavioural and Pharmacological Treatment for Tobacco Dependence in Pregnant Women with Complex Psychosocial Challenges: A Case Report"

_ijerph, 2020, doi:10.3390/ijerph17134770_

Round 1
Reviewer 1 Report
This is an interesting case study of intensive intervention to reduce cigarette use during pregnancy. The intensity of behavioral intervention used and the level of NRT available are quite significant in obtaining a fairly short term outcome. Its not clear what psychosocial support after delivery the patient received to assist with longer term abstention aside from the problem of cost of NRT outside of the pregnancy period. Given that this case study was conducted in the context of an existing iQuiP program, authors need to say more about what usual care for the program consisted of (line 134) and how unusual this case was in order to better understand its potential contribution to the literature about efficacy of interventions for tobacco cessation for pregnant women. The conclusions also seem to go beyond the case, especially since such contextual information about how out of the norm the case and the treatment intervention was. If the program is designed to serve similarly at risk women during pregnancy, some broader patterns might be drawn on concerning issues such as extent of support during and after pregnancy, and cost of NRT beyond pregnancy which might better support the conclusions of the manuscript than a single cases history.
Reviewer 2 Report
This is a case report in which a highly tobacco-dependent pregnant woman (23 weeks of gestation) with a history of substance use, mental illness, and trauma was enrolled in the Incentives to Quit Tobacco in Pregnancy programme. Treatment consisted of behavioural support, financial incentives, telephone-based counselling, and nicotine replacement therapy at doses above those recommended for pregnancy because of strong withdrawal symptoms.
Although nicotine replacement therapy in pregnancy is controversial because nicotine may produce foetal harm, it avoids other toxic substances contained in cigarettes, and therefore its use might be justified for heavy smokers even in pregnancy. Questionnaires were applied and showed that the participant had depression and anxiety symptoms; she had tried to quit smoking several times with no avail. She also suffered physical abuse from her partner and had a blood disorder, thus this was a very difficult case.
Even if financial incentives have been reported to be the most effective treatment for pregnant women to stop smoking, these incentives were provided only until delivery, while nicotine replacement therapy was provided until 12 weeks postpartum, when the programme ended. It was reported that the participant returned to cigarette smoking. The importance of providing incentives and counselling several weeks (or even months) after delivery is mentioned in the conclusion but it should be further discussed, even if nicotine-replacement therapy has ended, because the need of these incentives may motivate these women to continue the programme.
The participant was enrolled at week 23 of gestation, this is advanced pregnancy and irreversible effects caused by tobacco smoking and other substances could have been produced to the baby, therefore the importance of early enrolment should also be discussed.
Please indicate the correct age of the participant; the abstract (line 18) indicates 34 years old, while the text (line 73) indicates 30 years old, it is confusing.
In my opinion, the information provided in this manuscript is well presented and sufficient for a case report study.
Round 2
Reviewer 1 Report
The authors have largely been responsive to prior comments excerpt for one issue, that is what is the 'usual care' model for the program in which the case was enrolled. (See mention in line 134). Here would have been the place (or in the prior paragraph) to give the basic outline of the program-even though it may be covered in another manuscript. For example, line 146 indicates that sample packs of NRT were given after initial assessment but its not clear if it was just for this case, or its part of the usual care. So it would be helpful to clearly state what types of interventions this case required to be successful that were beyond the stated program model (ex behavioral support? financial incentives?). Or, rather, is the model flexible enough to provide a variety of intensity of services to achieve good outcomes (this case illustrating how much that was necessary), and therefore further investment in this particular model along with addressing the limitations of postpartum and longer term support would be important to improve maternal and child outcomes among women with multiple substance use and mental health issues.
I believe the points about the need for extensive support and continuing NRT and harm reduction interventions ongoing after delivery are important but its not quite clear if the program always ends at delivery (or 12 week post partum)? It seems there is an end date relative to the delivery date but the authors are recommending longer term follow up. Certainly a justification for longer term follow up is protecting the health of the newborn and other children from second hand tobacco smoke and/or the mother's other drug use which might be impacted by ongoing services for reduction or abstinence from drug and tobacco use.
